# LCSim: A Large-Scale Controllable Traffic Simulator

## Abstract

With the rapid growth of urban transportation and the continuous progress in autonomous driving, a demand for robust benchmarking autonomous driving algorithms has emerged, calling for accurate modeling of large-scale urban traffic scenarios with diverse vehicle driving styles. Traditional traffic simulators, such as SUMO, often depend on hand-crafted scenarios and rule-based models, where vehicle actions are limited to speed adjustment and lane changes, making it difficult for them to create realistic traffic environments. In recent years, real-world traffic scenario datasets have been developed alongside advancements in autonomous driving, facilitating the rise of data-driven simulators and learning-based simulation methods. However, current data-driven simulators are often restricted to replicating the traffic scenarios and driving styles within the datasets they rely on, limiting their ability to model multi-style driving behaviors observed in the real world. We propose *LCSim*, a large-scale controllable traffic simulator. First, we define a unified data format for traffic scenarios and provide tools to construct them from multiple data sources, enabling large-scale traffic simulation. Furthermore, we integrate a diffusion-based vehicle motion planner into LCSim to facilitate realistic and diverse vehicle modeling. Under specific guidance, this allows for the creation of traffic scenarios that reflect various driving styles. Leveraging these features, LCSim can provide large-scale, realistic, and controllable virtual traffic environments. Codes and demos are available at `https://anonymous.4open.science/r/LCSim-0C7A`.

## 1 Introduction

As global urbanization progresses, the complexity and diversity of urban transportation systems continue to increase. The driving styles of vehicles in various cities often have distinct characteristics (Sagberg et al., 2015), leading to high costs of benchmarking autonomous driving algorithms, which require high levels of safety and reliability and thorough testing and evaluation before actual deployment (Waymo, 2021). This necessitates accurately modeling urban microscopic traffic scenarios through traffic simulation, enabling the robust assessment of relevant algorithms. Accurate modeling of urban traffic scenarios poses two main challenges for simulation systems: the need for realistic and controllable vehicle models to replicate the complex and diverse driving behaviors in reality, and the requirement for large-scale traffic scenario data to support traffic simulation.

Existing simulation methods often have shortcomings in these two aspects. For one, vehicle behavior modeling in simulation systems is typically categorized into three types: rule-based (e.g. IDM (Brockfeld et al., 2003)) simulation (Behrisch et al., 2011; Li et al., 2022; Wenl et al., 2023; Zhang et al., 2019; Gulino et al., 2024; H. Caesar, 2021; Li et al., 2024; Liang et al., 2023; Zhang et al., 2024), log-replay of real-world dataset (Gulino et al., 2024; H. Caesar, 2021; Li et al., 2024), and learning-based methods (Bansal et al., 2018; Isele et al., 2018; Chai et al., 2019; Bergamini et al., 2021; Igl et al., 2022; Bronstein et al., 2022; Zhong et al., 2023). Rule-based simulations are often too simplistic and fail to replicate real vehicle behaviors accurately. Log replay and learning-based methods excel in replicating real vehicle behavior, but they usually lack controllability and struggle to model different driving styles faithfully. CTG (Zhong et al., 2023) has proposed a controllable traffic simulation method based on a diffusion model. But, its scenario is limited to the nuScenes (Caesar et al., 2020) dataset, and this method has not been integrated into a simulation system for algorithm benchmarking. On the other hand, most data-driven simulators rely on public datasets that

only contain fragmented scenarios (H. Caesar, 2021; Gulino et al., 2024; Li et al., 2024), limiting the scale of the simulation. Metadrive (Li et al., 2022) offers manual map creation tools. ScenarioNet (Li et al., 2024) does a great job of collecting large-scale real-world traffic scenarios from various driving datasets. However, the driving styles in the simulation environments they provide are still constrained by the given dataset. Therefore, extra efforts are needed to improve traffic scenario construction and enhance vehicle modeling.

We propose LCSim, a *Large-scale*, *Controllable traffic Simulator* to address the abovementioned challenges. Our contributions are listed below:

- We define a unified data format for traffic scenarios and provide tools to construct them from multiple data sources including real-world driving datasets like the Waymo open motion dataset (WOMD) (LLC, 2019) and Argoverse (Wilson et al., 2021) dataset, and hand-crafted scenarios built from public data sources such as OpenStreetMap (OSM)[1] (Behrisch et al., 2011; Zhang et al., 2024).

- We design and implement a simulation system that integrates a diffusion-based vehicle motion planner to achieve realistic, diverse, and controllable traffic simulation in the constructed traffic scenarios. A Gym-like environment interface is provided to support reinforcement learning algorithm training and benchmarking.

- A series of experiments are conducted to validate LCSim's functionality. Firstly, we demonstrate the ability of the diffusion-based motion planner on WOMD. Next, reinforcement learning agents are trained in environments with different driving styles built by LCSim, showcasing the impact of various driving styles on algorithm benchmarking. Lastly, through the accurate replication of city-level traffic scenarios, we highlight LCSim's capability to construct large-scale traffic simulations.

## 2 RELATED WORK

**Traffic Simulators.** The development of traffic simulators has a history of over a decade. Initially, researchers conducted simulations based on hand-crafted traffic scenarios and rule-based vehicle models (Behrisch et al., 2011; Dosovitskiy et al., 2017; Zhang et al., 2019; Liang et al., 2023; Wenl et al., 2023). However, these rule-based models are often simplistic and unable to accurately model real and diverse vehicle behaviors. With the continuous advancement of autonomous driving, an increasing number of open-source datasets containing real-world traffic scenarios have been released in recent years(LLC, 2019; Wilson et al., 2021; Caesar et al., 2020; H. Caesar, 2021; Houston et al., 2020). Consequently, many data-driven simulators based on these datasets have emerged (Kothari et al., 2021; Vinitsky et al., 2023; Li et al., 2024; Gulino et al., 2024). They utilize log replay to rebuild realistic traffic scenarios and incorporate rule-based models to enable close-looped simulations. Building upon this foundation, DriverGym (Kothari et al., 2021) provides a learning-based vehicle model based on SimNet (Bergamini et al., 2021), while ScenarioNet (Li et al., 2024) integrates various open-source datasets and offers reinforcement learning-based vehicle agent. However, these data-driven simulators often only simulate fragmented scenarios based on the provided data, and their simulation scale is limited only to the scope of the dataset. Metadrive (Li et al., 2022) presents a scenario-creation tool based on the combination of map elements. However, this approach faces challenges when it comes to constructing large-scale urban road networks. Furthermore, to the best of our knowledge, LCSim is the first open-source traffic simulator to provide controllable learning-based vehicle models to simulate multi-style driving behaviors in the real world.

**Learning-based Traffic Simulation.** Various learning-based vehicle simulation methods have emerged with the increasing availability of open-source traffic scenario datasets in recent years. Among them, imitation learning is often employed to learn expert actions from the dataset, thereby achieving realistic traffic simulation (Xu et al., 2023; Bergamini et al., 2021; Bhattacharyya et al., 2018; Zheng et al., 2020; Bhattacharyya et al., 2022; Yan et al., 2023). However, this approach is often plagued by causal confusion (De Haan et al., 2019) and distribution shift (Ross et al., 2011). Reinforcement learning methods, on the other hand, address the distribution shift issue effectively by interacting with the simulation environment to learn driving behaviors (Kendall et al., 2019; Isele et al., 2018; Lu et al., 2023; Wang et al., 2018; Zheng et al., 2022). However, the design of reward

---

[1]https://www.openstreetmap.org/

Table 1: Comparison of related traffic simulators. LCSim provides automated tools for traffic scenario construction and diffusion-based controllable vehicle motion planning.

| | Scenario Construction | RL Environment | Rule-based Agent | Data-driven Agent | Controllable | Sensor Sim |
|---|---|---|---|---|---|---|
| SUMO | ✔ | | ✔ | | ✔ | |
| nuPlan-devkit | | | ✔ | | | ✔ |
| DriverGym | | ✔ | ✔ | ✔ | | |
| MetaDrive | ✔ | ✔ | ✔ | ✔ | | ✔ |
| Waymax | | ✔ | ✔ | | | |
| TrafficSim | | | | ✔ | | |
| SimNet | | | | ✔ | | |
| CTG | | | | ✔ | ✔ | |
| **LCSim (ours)** | ✔ | ✔ | ✔ | ✔ | ✔ | |

functions and the construction of the simulation environment are often complex. As generative models have advanced, many researchers have started to utilize the generation of vehicle motion plans for simulation purposes (Suo et al., 2021; Tan et al., 2021; Zhang et al., 2023; Rempe et al., 2022; Tang et al., 2021; Krajewski et al., 2018; Zhong et al., 2023). Among these approaches, CTG (Zhong et al., 2023) utilizes a diffusion model to achieve controllable vehicle simulation. However, this method has not been used in traffic simulators for further algorithm training and testing. Following their idea, we trained our diffusion-based vehicle motion planner based on WOMD. Controllable traffic simulation can be achieved by generating vehicle motion plans with guide functions.

## 3 SYSTEM DESIGN

### 3.1 SCENARIO DATA CONSTRUCTION

To achieve large-scale traffic simulation, we define a unified data format based on Protobuf for traffic scenario data from multiple data sources including real-world driving datasets and hand-crafted scenarios provided by MOSS (Zhang et al., 2024) toolchain.

**Unified Scenario Data Format.** Figure 1 shows an example of our unified scenario data format. Each scenario data consists of the following three parts:

- Map: The map data consists of three components: lanes, roads, and junctions. The lanes contain the primary map information, with each lane element storing the lane's ID, type, polygon information of the centerline, and the connections between the current lane and others (such as predecessors, successors, and neighbors). Roads and junctions serve as containers for lane elements, both containing boundary information for the drivable areas on the map. Each lane belongs to a unique road or junction. Additionally, junctions store information related to traffic lights, including the IDs of the lanes they control and the traffic signal phases.

- Agents: The Agents data contains all the information about agents that need simulating. Each agent element includes the agent's ID, attributes, and routes. The attributes consist of basic properties such as the agent's type and shape, while the routes contain the full travel schedules for the agent. An agent's schedule data includes the departure time and reference route,

```
{ "Map": {
  "lanes": [
    { "id": 0,
      "type": "surface_street",
      "center_line": [{x,y,z}],
      "connections": [{n,p,s}]
    }, ...
  ],
  "roads": [
    { "lane_ids": [0,1,2],
      "boundaries": [{x,y,z}]
    }, ...
  ],
  "junctions": [
    { "lane_ids": [3,4,5],
      "boundaries": [{x,y,z}],
      "traffic_light": {
        "control_ids": [3,4,5],
        "phases": [{r,b,y}]
      }
    }, ...
  ]
},
"Agents": [
  { "id": 0,
    "attributes": {
      "type": "vehicle",
      "shape": [l,w,h]
    },
    "routes": [{t,[{x,y,z}]}]
  }, ...
]}
```

Figure 1: The unified format of traffic scenario data.

with the reference route detailing the agent's state (position, speed, heading, etc.) at each step after the departure time. In log replay-based simulations, this information is provided by scenario data, whereas in diffusion-based simulations, it is generated by the diffusion model and continuously updated throughout the simulation.

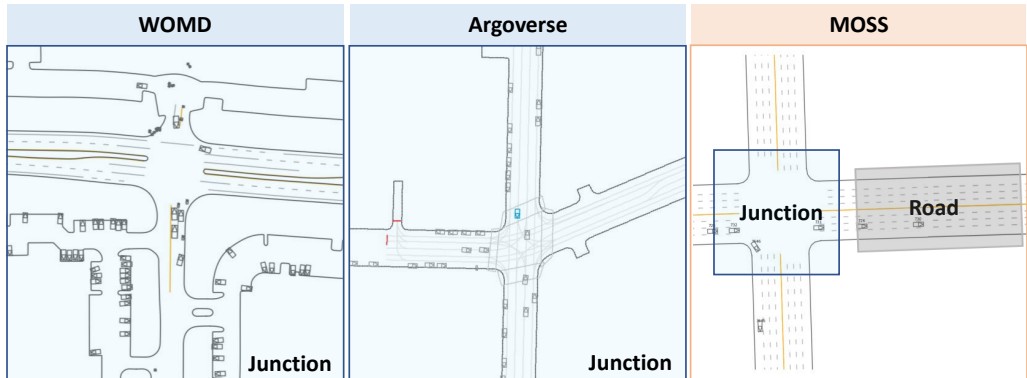

Figure 2: Traffic scenarios from different data sources.

**Scenario Construction.** Figure 2 illustrates traffic scenarios constructed from different data sources. For the WOMD and Argoverse datasets, all map information is placed within a single junction. We aligned the basic attributes of the map features based on the map element types provided in WOMD. For the scenarios obtained from the MOSS toolchain, we completed the map elements, including drivable boundaries and lane lines as the original data only contains centerlines. In the MOSS scenario, agent routes are provided as origin-destination points, and we implemented an A-star-based router to complete them into full reference routes. MOSS allows the construction of arbitrarily large scenarios using latitude and longitude ranges. We further divide the map into roads and junctions, and during the simulation, the map elements and agents in each road or junction are organized into a data instance. These instances are processed in batches by the diffusion model for vehicle motion planning, allowing users to select areas of interest by ID and enable diffusion-based simulation within those areas. Details of this part can be found in Appendix B.1.

## 3.2 SIMULATION ARCHITECTURE

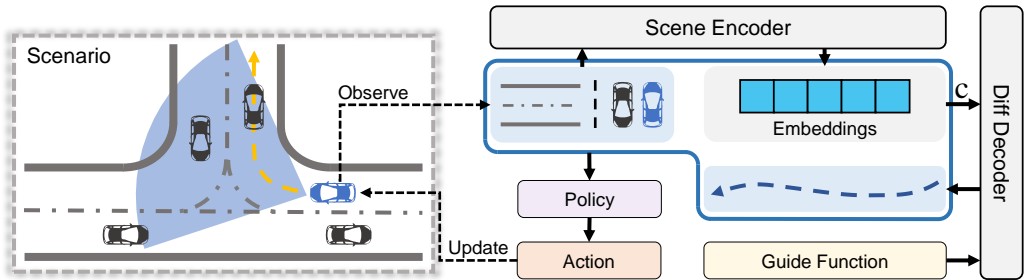

Figure 3: The simulation architecture of LCSim.

With scenario data constructed from multiple sources, LCSim performs discrete-time simulation based on a given time interval. Figure 3 illustrates the basic components of a simulation step. Each simulation step can be divided into two stages:

**Prepare Stage.** During this stage, the simulator prepares the observation data for each vehicle, as depicted in the blue box in Figure 3. The observation data comprises three components: scene information observed by the vehicle, including road network topology and surrounding vehicles, scene embeddings computed by the scene encoder, which encodes the scene information, and vehicles' motion plans either generated by the diffusion decoder or given by the logs from the scenario data.

**Update Stage.** In this stage, each vehicle's action is calculated by its policy based on the observation data, and these actions are used to update the vehicles' states. We implement various control policies in the simulator to handle different simulation scenarios. The *ExpertPolicy* controls vehicles to strictly follow the given motion plans. At the same time, the *BicycleExpertPolicy* enhances this by adding kinematic control based on the bicycle model to achieve more realistic simulation effects. Furthermore, we implement the *IDMPolicy* to enable closed-loop traffic simulation, where vehicles adjust their accelerations based on the objects ahead while following the given motion plans. Additionally, any vehicle in the scenario can be controlled by external input actions, allowing the simulator to serve as a training or testing environment for specific vehicle control algorithms. Details about these policies can be found in Appendix B.2.

### 3.3 DIFFUSION-BASED MOTION PLANNER

We design and implement a vehicle motion planning module based on a diffusion model to achieve controllable traffic simulation. During the simulation process, this module takes the scene information of the current step and a guidie function as inputs to generate realistic and controllable motion plans for vehicles in the scenario. Algorithm 1 summarizes the guided generation process of the model. The entire model consists of the following three main components:

**Scene Encoder.** For accurately modeling the behavior of traffic participants, feature representations of scene information including map elements and historical states of traffic participants are required as conditions for the diffusion model. Following (Zhou et al., 2023a; Shi et al., 2023), we utilize a spatial-temporal attention mechanism to model the scene features, taking in map polygons and historical states of agents to compute scene embeddings for each vehicle in the scenario.

---

**Algorithm 1** Generate Controllable Motion Plans

1: **Require** diffusion decoder $D_{\boldsymbol{\theta}}$, scene embeddings $\boldsymbol{c}$, guide function $\mathcal{G}$, diffusion steps $t_{i \in \{0,\ldots,N\}}$, guide gradient descent steps $K$, guide scale $\alpha$, guide clip $\beta$, initial noise level $S_{noise}$
2: Initialize white noise $\boldsymbol{\tau}^0 \sim \mathcal{N}(\boldsymbol{0}, S_{noise}^2 \boldsymbol{I})$
3: **for** $i = 0,\ldots,N$ **do**
4:     *Denoising Step*
5:     $\boldsymbol{d}_i = (\boldsymbol{\tau}^i - D_{\boldsymbol{\theta}}(\boldsymbol{\tau}^i; \boldsymbol{c}, t_i))/t_i$
6:     $\boldsymbol{\tau}^{i+1} = \boldsymbol{\tau}^i + (t_{i+1} - t_i)\boldsymbol{d}_i$
7:     **if** $t_{i+1} \neq 0$ **then**
8:         $\boldsymbol{d}_i' = (\boldsymbol{\tau}^{i+1} - D_{\boldsymbol{\theta}}(\boldsymbol{\tau}^{i+1}; \boldsymbol{c}, t_{i+1}))/t_{i+1}$
9:         $\boldsymbol{\tau}^{i+1} = \boldsymbol{\tau}^i + (t_{i+1} - t_i)(\frac{1}{2}\boldsymbol{d}_i + \frac{1}{2}\boldsymbol{d}_i')$
10:    *Guide Step*
11:    $\boldsymbol{\tau}_0^{i+1} = \boldsymbol{\tau}^{i+1}$
12:    **for** $j = 1,\ldots,K$ **do**
13:       $\boldsymbol{\tau}_j^{i+1} = \boldsymbol{\tau}_{j-1}^{i+1} + \alpha\nabla\mathcal{G}(\boldsymbol{\tau}_{j-1}^{i+1})$
14:       $\Delta\boldsymbol{\tau} = |\boldsymbol{\tau}_j^{i+1} - \boldsymbol{\tau}_0^{i+1}|; \Delta\boldsymbol{\tau} \leftarrow clip(\Delta\boldsymbol{\tau}, -\beta, \beta)$
15:       $\boldsymbol{\tau}_j^{i+1} \leftarrow \boldsymbol{\tau}_0^{i+1} + \Delta\boldsymbol{\tau}$
16: Execute motion plans of each vehicle using output $\boldsymbol{\tau}_K^N$

---

**Denoising Process.** The vehicle's future velocities and heading angles are used as the generation target for the diffusion model. Like QCNet (Zhou et al., 2023a), we employ an attention-based architecture for the diffusion decoder. The decoder takes the noised input data combined with the noise level as query values. It performs cross-attention between input queries and the scene embeddings, resulting in denoised data as the output. The training and sampling process of the diffusion model follows Nvidia's EDM architecture (Karras et al., 2022).

**Guide Function.** Similar to CTG (Zhong et al., 2023), we impose a loss function on the intermediate results of the denoising process and backpropagate the gradients to guide the generation process of the diffusion model. In our experiments, the control targets include realistic guidance, such as no collision and staying on the road, as well as vehicle behavior style guidance, which encompasses factors like max acceleration, target speed, and time headway. Furthermore, by guiding surrounding vehicles to approach the target vehicle, a high-collision-rate adversarial driving environment can be produced for the target vehicle.

Our diffusion model can generate vehicle motion plans for 8 seconds in the future, we employ a recurrent generation approach based on a specified time interval during the simulation. More details about the model design can be found in Appendix A.

## 4 EXPERIMENTS

We conduct a series of experiments to validate LCSim's functionality. Firstly, we demonstrate how the diffusion-based motion planner in LCSim is constructed and its ability to create realistic and controllable traffic scenarios. Next, we train reinforcement learning agents in simulation environments with different driving styles built by LCSim, showcasing the impact of various driving environments

Table 2: Evaluation results of our diffusion-based motion planner.

| | Collision (%) | Off-Road (%) | minADE (m) | minFDE (m) |
|---|---|---|---|---|
| TrafficSim | 4.901 (± 0.019) | 2.034 (± 0.021) | 1.205 (± 0.001) | 3.267 (± 0.027) |
| SimNet | 5.011 (± 0.013) | 1.996 (± 0.017) | **1.201** (± 0.001) | 3.259 (± 0.025) |
| Ours w/o guide | 9.693 (± 0.413) | 2.901 (± 0.019) | 1.383 (± 0.002) | **2.869** (± 0.005) |
| Ours | **4.118** (± 0.082) | **1.521** (± 0.110) | 1.526 (± 0.005) | 3.077 (± 0.034) |

on agent performance. Lastly, through the accurate replication of city-level traffic scenarios, we highlight LCSim's capability to construct large-scale traffic simulations.

## 4.1 PERFORMANCE OF DIFFUSION-BASED MOTION PLANNER

**Datasets.** We train our diffusion model on WOMD (LLC, 2019), which contains 500+ hours of driving logs collected from seven different cities in the United States. The dataset is further divided into scenario segments of 20s and 9s. In this experiment, we utilize the 9s segments for training, using the initial 1s as the historical context, and let the model generate vehicle motions in the future 8s. The diffusion model is trained on the training set and evaluated for its performance on the validation set.

**Metrics.** The evaluation metrics of the model consist of two aspects: first, the motion plans generated for vehicles by the model should adhere to basic traffic rules. We calculate the probabilities of vehicle collisions and off-road incidents in the simulated scenarios and compare these with statistical values from real-world data. Second, we use distance-based metrics to assess the model's ability to reconstruct real-world traffic scenarios. For each scenario in the validation set, we sample K times for a simulation duration of T and compute the average displacement error across time (ADE) and the final displacement error (FDE) at the last time step. In the experiment, we set K = 6, and T = 8s, and choose the best matching sample to calculate minADE and minFDE.

**Baselines & Settings.** To validate the effectiveness of our model, we compare it with two famous learning-based traffic simulation methods, TrafficSim (Suo et al., 2021) and SimNet (Bergamini et al., 2021). SimNet simulates traffic based on behavior cloning, while TrafficSim generates motion plans for vehicles using C-VAE. Additionally, to verify the effectiveness of the realistic guidance, we compare the model's performance with and without the "no collision" and "no off-road" guidance [2]. To handle the impact of randomness in the generation process, we conducted repeated experiments and reported the mean and error bars of the results.

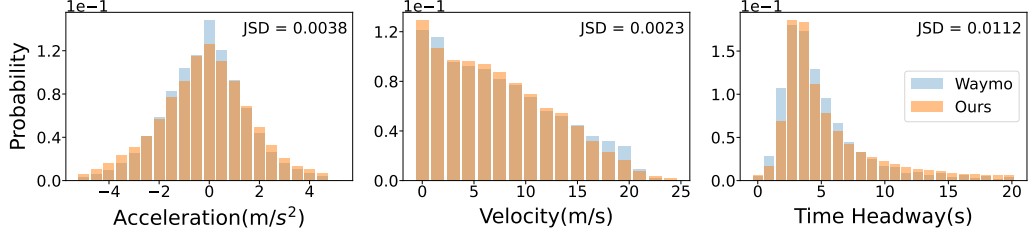

Figure 4: The comparison of vehicle behaviors between WOMD and our diffusion-based simulation.

**Results.** Table 2 presents the quantitative results of the experiments. While our method lags behind the baseline in terms of accurately imitating vehicle trajectories in the dataset, it surpasses the baseline in adherence to traffic rules, realistic vehicle interactions, and the accuracy of long-term simulations. Additionally, the comparison between simulations with and without realistic guidance demonstrates the effectiveness of the realistic guide functions. Furthermore, Figure 4 shows a comparison of vehicle behavior distributions collected from the original dataset and our diffusion-based simulations, indicating that our model can learn the driving styles present in the dataset. Visualization of the generating process can be seen in Appendix A.

---

[2]See Appendix A.4 for details.

## 4.2 ALIGNMENT OF VEHICLE BEHAVIOR CHARACTERISTICS

**Private Driving Dataset.** Our private driving dataset comprises about 400 hours of vehicle driving logs collected from vehicles in the Beijing Yizhuang area. The data is presented in a format similar to vehicle trajectories in WOMD. We conducted statistical analysis on the dataset, focusing on metrics such as acceleration, relative distance, and time headway during the car following process. This analysis allowed us to derive the driving behavior characteristics of vehicles in the Yizhuang area. Details about the dataset can be found in Appendix C.

In Figure 5, we compare the behavioral characteristics of vehicles in our private dataset with those in WOMD. The comparison includes metrics such as vehicle acceleration, relative distance, and time headway during car following. Since a vehicle's driving speed is often related to the specific driving environment (e.g., road congestion, lane speed limits) rather than the behavioral characteristics, we do not include speed in the comparison. It can be observed that there are significant differences between the behaviors of vehicles in these two datasets. Vehicles in the Yizhuang area exhibit a more "gentle" driving style, showing a preference for using smaller accelerations during start and brake. Additionally, they maintain larger relative distances and headway times during the following process compared to vehicles in the Waymo dataset.

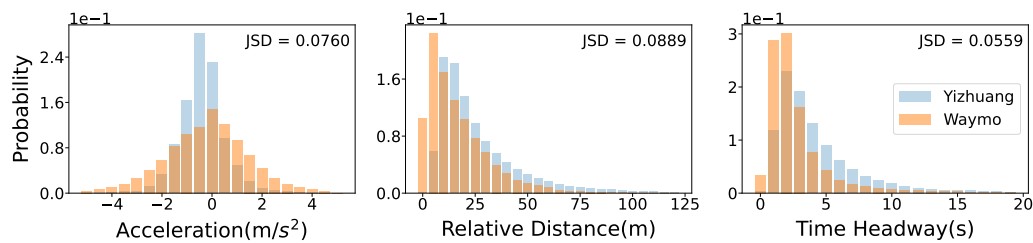

Figure 5: The differences in vehicle behavior between WOMD and private datasets.

As our model is trained on WOMD, without imposing any guidance on the generation process, the vehicle behavior characteristics in the diffusion-based simulation remain consistent with the Waymo dataset, as shown in Figure 4. By applying guide functions including max acceleration, relative distance, and time headway during the generation process, we can align the vehicle behavior characteristics produced by the diffusion-based simulation with those collected in our private driving dataset. The comparison of the two distributions can be seen in Figure 6. This demonstrates that our simulator can model vehicles with diverse driving styles, thereby providing traffic simulation environments with different driving styles. Details about the guide function we use here can be found in Appendix A.4.

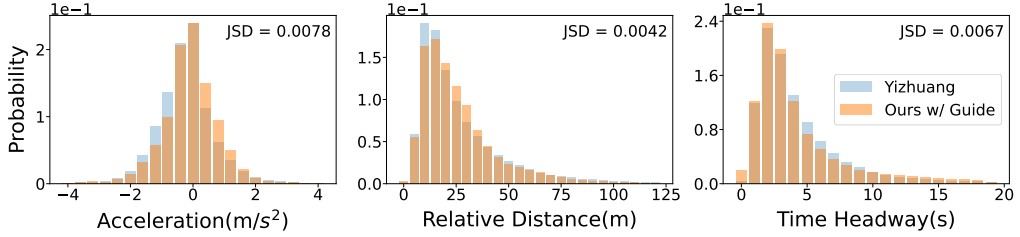

Figure 6: The comparison of vehicle behaviors between the private dataset and guided diffusion-based simulation.

## 4.3 MULTI-STYLE RL TRAINING

We construct a single-agent reinforcement learning environment based on WOMD with our guided diffusion-based simulation to further investigate the impact of traffic environments with different driving styles on driving policy learning. Below are the training settings and results:

Table 3: Evaluation results of RL agents.

| | Collision Rate (%) | Off-Road Rate (%) | Route Progress (%) | Success Rate (%) | Reward |
|---|---|---|---|---|---|
| WOMD | 15.71 (± 0.56) | 4.24 (± 0.20) | **84.98** (± 0.53) | 51.31 (± 0.56) | 7.41 (± 0.10) |
| Diff w/ gentle | 32.28 (± 0.52) | 3.52 (± 0.08) | 59.06 (± 0.42) | 21.52 (± 0.18) | 0.50 (± 0.06) |
| Diff w/ adv | **8.72** (± 0.23) | 15.49 (± 0.42) | 83.88 (± 0.43) | 33.53 (± 0.27) | 5.05 (± 0.05) |
| Diff w/o guide | 12.06 (± 0.28) | **3.01** (± 0.12) | 84.76 (± 0.35) | **52.75** (± 0.50) | **8.59** (± 0.14) |

**Settings.** To validate the model's capability in unseen scenarios, we construct a reinforcement learning environment based on the validation set of WOMD. We select 4,400 scenarios from the validation set and further divide them into a training set containing 4,000 scenarios and a test set containing 400 scenarios. We train a PPO (Schulman et al., 2017) agent on the training set and evaluate its performance on the test set. As shown in Figure 7, we let the PPO agent control the self-driving car (SDC) marked in WOMD, the agent's observation space consists of scene embedding computed by the scene encoder and a reference route for the vehicle. In different training environments, the route is either given by driving logs from the dataset or computed by motion plans generated by the diffusion model. The background vehicles are controlled by policies within our simulator. On the test set, we test four metrics of the agent: collision rate, off-road rate, average route progress rate, and scenario success rate. We also provide the average reward value per episode. The background vehicles of the test set act based on WOMD driving logs. Detailed RL training settings can be found in Appendix D.1.

**Styles of the Training Environments.** We create four distinct driving environments on the training set: In the first one, vehicles base their actions on real trajectories from WOMD driving logs. The second one utilizes the diffusion model without guide functions, which maintains consistency with WOMD in terms of vehicle behavior styles. With the diffusion model's nature, it generates diverse vehicle motions under the same initial conditions, exposing the agent to a broader range of traffic scenarios during training. The third one follows the driving style observed in our private driving dataset, emphasizing a more "gentle" driving behavior compared to the WOMD-based environment. Furthermore, an adversarial driving environment is implemented by guiding nearby vehicles closer to the agent, creating a training scenario with a higher potential for collisions. Details about the configuration are available in Appendix D.2.

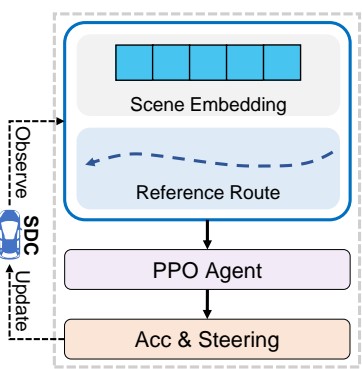

Figure 7: Workflow of the RL agent.

**Results.** Table 3 presents the performance of agents trained in environments with different driving styles on the test set. Compared to agents trained on the original WOMD driving logs, those trained in diffusion-based simulation environments without guidance perform better across almost all metrics. This improvement is attributed to the diffusion-based simulations increasing the diversity of traffic scenarios in the training environment, enabling agents to learn more general and effective driving strategies. Conversely, agents trained in "gentler" environments perform poorly on the test set, as differences in background vehicle behavior between the training and test sets result in the agent's driving strategies being ineffective in avoiding collisions. Additionally, the more passive driving style in the training environment's reference routes leads to a lower route progress rate in the test set. Agents trained in adversarial environments excel at avoiding collisions, but their maneuvers to evade surrounding vehicles also result in a higher probability of driving off the road.

## 4.4 CITY-SCALE TRAFFIC SCENARIO CONSTRUCTION

We showcase the scalability of LCSim with city-scale simulations of real-world traffic scenarios in two metropolises.

**Datasets.** We use two vehicle trajectory datasets (Yu et al., 2023; 2022; Lin et al., 2021) each with one-day-long city-scale trajectories of the entire fleet recovered from daily urban traffic camera

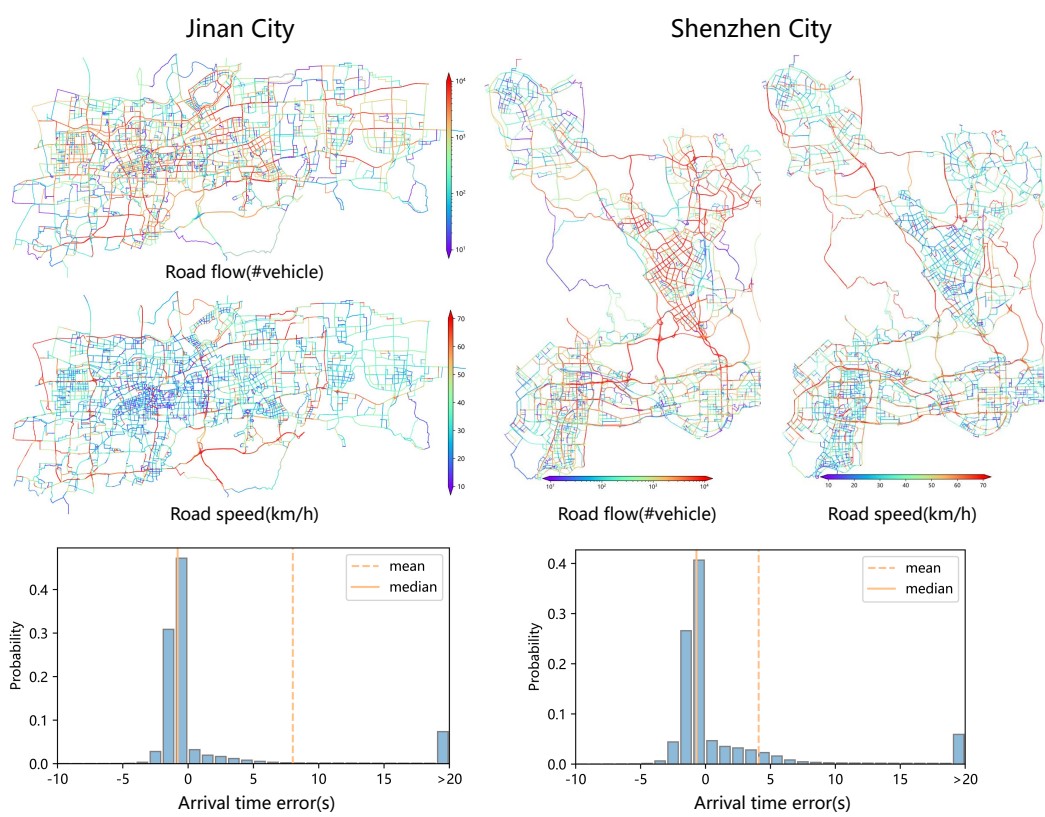

Figure 8: City-scale traffic scenario simulations in Jinan and Shenzhen.

videos in Jinan and Shenzhen city. Both of the datasets involve over one million trajectories and one thousand square kilometers of urban area.

**Settings.** Compared with open-source driving datasets like WOMD, the trajectories recovered from traffic cameras are temporally sparser, where only the arrival time at road intersection is specified (Yu et al., 2023), which is thus taken as the travel schedule of each agent in LCSim with a series of trips between intersections with corresponding departure time and arrival time. We show that by simulating the vehicles given their schedules, using the arrival time at a specific position as goal point guidance [3], LCSim can effectively replicate real-world city-scale traffic scenarios.

**Results.** Figure 8 shows, in Jinan and Shenzhen, the spatial distribution of simulated road flow and speed, as well as the probability distribution of the arrival time error which is the deviation of the simulated arrival time of each trip compared with that of the ground truth trajectory data. As can be seen, the arrival time error is mainly distributed around zero with over 90% of trips having arrival time errors less than 20 seconds. LCSim also produces reasonable traffic conditions with coherence between the road network structure, flow, and speed.

## 5 CONCLUSION

We proposed LCSim, a large-scale, controllable diffusion-based traffic simulator. With an automated tool to construct traffic scenarios from multiple data sources, LCSim is capable of conducting large-scale traffic simulations. By integrating the diffusion-based motion planner and guide functions, LCSim can build traffic environments with diverse vehicle driving styles.

**Limitations.** LCSim has two main limitations. Firstly, the simulator is implemented in Python using a single-threaded CPU, which limits its performance potential, discussion can be found in Appendix

---

[3]See Appendix A.4 for details.

E. Although parallel simulation using multiple processes is currently employed as a solution, it does not fundamentally address the issue. One potential approach to overcome this limitation is to develop a multi-threaded version of the simulator using C++ and deploy the Diffusion model in C++, which is a potential direction for future work. Secondly, the simulator currently provides visualization only from a top-down perspective and lacks the rendering of realistic perceptual data. Integrating the sensor data generation method into the simulation is one of the planned future developments to address this limitation.

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

## APPENDIX

In the appendix, we provide more details about the experiments discussed in the main text. Section A introduces the implementation details of the diffusion model and the specific content and form of the guide function. Section B details the implementation of the system and showcases the visualization of scenarios in the simulator. Section C covers the relevant content of our private driving dataset, while Section D delves into the detailed experimental configurations for multi-style reinforcement learning experiments. Section E discusses the efficiency issue of our simulator. Code and demos are available at https://anonymous.4open.science/r/LCSim-0C7A.

## A    DIFFUSION MODEL

Figure 9 shows the diffusion denoising process. With the road network topology and vehicle historical states as input, the model generates future vehicle motion plans through a denoising diffusion process.

Due to the relevant regulations of the Waymo Open Motion Dataset (WOMD) (LLC, 2019), we cannot provide the parameters of the model trained on it. In this section, we introduce the implementation details of the diffusion model and the hyperparameters used for training and inference in detail to ensure that the relevant experimental results can be easily reproduced.

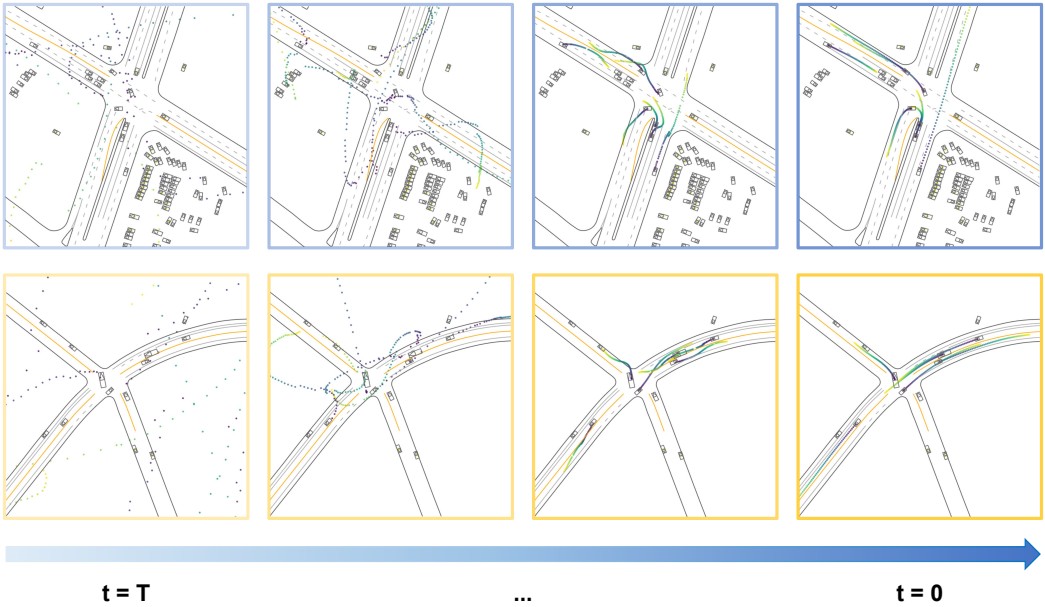

t = T                    ...                    t = 0

Figure 9: The process of generating vehicle action sequences by diffusion model.

### A.1    PROBLEM FORMULATION

Similar to (Li et al., 2024), we denote a traffic scenario as $\omega = (M, A_{1:T})$, where $M$ contains the information of a High-Definition (HD) map and $A_{1:T} = [A_1, ..., A_T]$ is the state sequence of all traffic participates. Each element $m_i$ of $M = \{m^1, ..., m^{N_m}\}$ represents the map factor like road lines, road edges, centerline of lanes, etc. And each element $a_i^t$ of $A_t = \{a_t^1, ..., a_t^{N_a}\}$ represents the state of the ith traffic participate at time step t including position, velocity, heading, etc.

Given the map elements $M = \{m^1, ..., m^{N_m}\}$ and the historical states of agents $A_{t_c-T_h:t_c}$, where $T_h$ is the number historical steps and $0 < T_h < t_c$, the model generates the future states of agents in the scenario $A_{t_c:t_c+T_f}$, where $T_f$ is the number of future steps.

Table 4: The attention mechanisms of scene encoder.

|  | Query | Key | Value |
|---|---|---|---|
| Agent Temporal | $\mathbf{v}^a_{i,t_c}$ | $\mathbf{v}^a_{i,t}$ | $\mathbf{v}^a_{i,t} \oplus Pos(t - t_c)$ |
| Agent-Map | $\mathbf{v}^a_{i,t_c}$ | $\mathbf{v}^m_j$ | $\mathbf{v}^m_j \oplus \mathbf{e}^{a\rightarrow m}_{ij}$ |
| Agent-Agent | $\mathbf{v}^a_{i,t_c}$ | $\mathbf{v}^a_{j,t_c}$ | $\mathbf{v}^a_{j,t_c} \oplus \mathbf{e}^{a\rightarrow a}_{ij}$ |

## A.2 MODEL ARCHITECTURE

**Scene Encoder.** We implemented our scene encoder based on MTR (Shi et al., 2023) and QCNet (Zhou et al., 2023a). As mentioned before, at each time step $t_c$, the input to the scene encoder includes the map elements $M = \{m^1, ..., m^{N_m}\}$ and the historical states of agents $A_{t_c-T_h:t_c}$. First, we construct a heterogeneous graph $G = (V, E)$ based on the geometric relationships among input features. The node set $V$ contains two kinds of node $v^a$ and $v^m$ and the edge set $E$ consists of three kinds of edge $e^{a\rightarrow a}$, $e^{a\rightarrow m}$ and $e^{m\rightarrow m}$. Connectivity is established between nodes within a certain range of relative distances. For nodes like $v^a_i$ and $v^m_j$, their node features contain attributes independent of geographical location like lane type, agent type, agent velocity, etc. The position information of nodes is stored in the relative form within the edge features like $e^{a\rightarrow m}_{ij} = [\mathbf{p}^m_j - \mathbf{p}^a_i, \theta^m_j - \theta^a_i]$, where $\mathbf{p}$ and $\theta$ are position vector and heading angle of each node at current time step $t_c$. For each category of elements in the graph, we use an MLP to map their features into the latent space with dimension $N_h$ to get the node embedding $\mathbf{v}^a_{i,t}(t_c - T_h \leq t \leq t_c)$, $\mathbf{v}^m_j$ and edge embedding $\mathbf{e}^{a\rightarrow a}_{ij}, \mathbf{e}^{a\rightarrow m}_{ij}, \mathbf{e}^{m\rightarrow m}_{ij}$. Then we apply four attention mechanisms in Table 4 to them to get the final scene embedding. The scene embedding consists of two components: the map embedding with a shape of $[M, N_h]$, and the agent embedding with a shape of $[A, T_h, N_h]$.

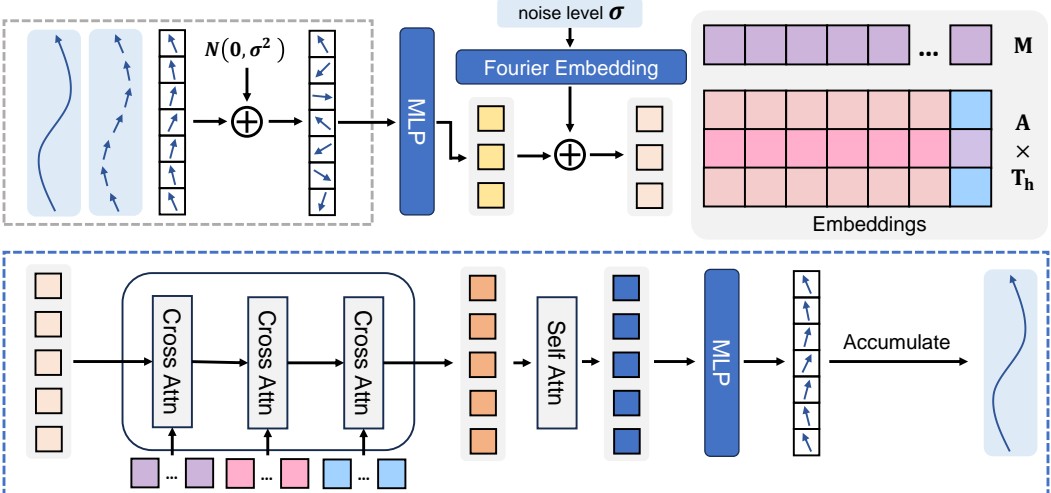

Figure 10: The architecture of diffusion decoder.

**Diffusion Decoder.** Figure 10 shows the whole architecture of the diffusion decoder. Similar to (Zhou et al., 2023b), we implemented a DETR-like decoder to model the joint distribution of multi-agent action sequences. Denote the generation target as $\boldsymbol{x} \in \mathbb{R}^{A\times T_f \times N_a}$, which represents future $T_f$ steps' actions of agents in the scenario. Firstly, noise $\boldsymbol{z} \sim \mathcal{N}(\mathbf{0}, \sigma^2)$ is added to the input sequence. Subsequently, the action sequence with noise for each agent is mapped to a latent space via an MLP, serving as the query embedding for that agent. The query is then added to the Fourier Embedding with noise level $\sigma$, similar to positional encoding, to inform the model about the current noise level. Next, the query vector undergoes cross-attention with map embeddings, embeddings of other agents in the scenario, and the historical state embedding of the current agent, resulting in a fused agent feature vector incorporating environmental information. Following this, self-attention is applied to

Table 5: Model parameters

| Parameter | Value |
|---|---|
| Input Size | 2 |
| Output Size | 2 |
| Embedding Size | 128 |
| Num Historical Steps | 10 |
| Num Future Steps | 80 |
| Num Polygon Types | 20 |
| Num Freq Bands | 64 |
| Map Encoder | |
|    Hidden Dim | 64 |
|    Num Layers | 5 |
|    Num Pre Layers | 3 |
| Agent Encoder | |
|    Time Span | 10 |
|    a2a Radius | 50 |
|    pl2a Radius | 50 |
|    Num Layers | 2 |
|    Num Heads | 8 |
|    Head Dim | 64 |
|    Dropout | 0.1 |
| Diff Decoder | |
|    Output Head | False |
|    Num t2m Steps | 10 |
|    pl2m Radius | 150 |
|    a2m Radius | 150 |
|    Num Layers | 2 |
|    Num Recurrent Steps | 2 |
|    Num Heads | 8 |
|    Head Dim | 64 |
|    Dropout | 0.1 |

Table 6: Training parameters

| Parameter | Value |
|---|---|
| Batch Size | 16 |
| Num Epochs | 200 |
| Weight Decay | 0.03 |
| Learning Rate | 0.0005 |
| Learning Rate Schedule | OneCycleLR |
| $\sigma_{data}$ | 0.1 |
| $c_{in}(\sigma)$ | $1/\sqrt{\sigma^2 + \sigma_{data}^2}$ |
| $c_{skip}(\sigma)$ | $\sigma_{data}^2/(\sigma^2 + \sigma_{data}^2)$ |
| $c_{out}(\sigma)$ | $\sigma \cdot \sigma_{data}/\sqrt{\sigma^2 + \sigma_{data}^2}$ |
| $c_{noise}(\sigma)$ | $\frac{1}{4}\ln\sigma$ |
| Noise Distribution | $\ln(\sigma) \sim \mathcal{N}\left(P_{\text{mean}}, P_{\text{std}}^2\right)$ |
| $P_{\text{mean}}$ | -1.2 |
| $P_{\text{std}}$ | 1.2 |

the feature vectors of each agent to ensure the authenticity of interaction among the action sequences generated for each agent. Finally, the feature vectors from the latent space are mapped back to the agent's action space via an MLP to obtain the de-noised agent action sequence.

## A.3 TRAINING DETAILS

**Training Target.** Diffusion model estimates the distribution of generation target $\boldsymbol{x} \sim p(\boldsymbol{x})$ by sampling from $p_{\boldsymbol{\theta}}(\boldsymbol{x})$ with learnable model parameter $\boldsymbol{\theta}$. Normally we have $p_{\boldsymbol{\theta}}(\boldsymbol{x}) = \frac{-f_{\boldsymbol{\theta}}(\boldsymbol{x})}{Z_{\boldsymbol{\theta}}}$, and use max-likelihood $\max_{\boldsymbol{\theta}} \sum_{i=1}^{N} \log p_{\boldsymbol{\theta}}(\boldsymbol{x}_i)$ to get parameter $\boldsymbol{\theta}$. However, to make the max likelihood training feasible, we need to know the normalization constant $Z_{\boldsymbol{\theta}}$, and either computing or approximating it would be a rather computationally expensive process, So we choose to model the score function $\nabla_{\boldsymbol{x}} \log p_{\boldsymbol{\theta}}(\boldsymbol{x}; \sigma)$ rather than directly model the probability density, with the score function, one can get data sample $\boldsymbol{x}_0 \sim p_{\boldsymbol{\theta}}(\boldsymbol{x})$ by the following equation (Jiang et al., 2023):

$$\boldsymbol{x}_0 = \boldsymbol{x}(T) + \int_T^0 -\dot{\sigma}(t)\sigma(t)\nabla_{\boldsymbol{x}} \log p_{\boldsymbol{\theta}}(\boldsymbol{x}(t); \sigma(t))dt \quad \text{where } \boldsymbol{x}(T) \sim \mathcal{N}\left(\boldsymbol{0}, \sigma_{\max}^2 \boldsymbol{I}\right) \quad (1)$$

On this basis, we add a condition $\boldsymbol{c}$ composed of scene embeddings and use our model to approximate the score function $\nabla_{\boldsymbol{x}} \log p_{\boldsymbol{\theta}}(\boldsymbol{x}; \boldsymbol{c}, \sigma) \approx (D_{\boldsymbol{\theta}}(\boldsymbol{x}; \boldsymbol{c}, \sigma) - \boldsymbol{x})/\sigma^2$, which leads to the training target (Jiang et al., 2023):

$$\mathbb{E}_{\boldsymbol{x}, \boldsymbol{c} \sim \chi_c} \mathbb{E}_{\sigma \sim q(\sigma)} \mathbb{E}_{\boldsymbol{\epsilon} \sim \mathcal{N}(\boldsymbol{0}, \sigma^2 \boldsymbol{I})} \|D_{\boldsymbol{\theta}}(\boldsymbol{x} + \boldsymbol{\epsilon}; \boldsymbol{c}, \sigma) - \boldsymbol{x}\|_2^2 \quad (2)$$

$\chi_c$ is the training dataset combined with embeddings computed by the scene encoder, and $q(\sigma)$ represents the schedule of the noise level added to the original data sample. For better performance,

we introduce the precondition as described in (Karras et al., 2022) to ensure that the input and output of the model both follow a standard normal distribution with unit variance:

$$D_{\boldsymbol{\theta}}(\boldsymbol{x}; \boldsymbol{c}, \sigma) = c_{\text{skip}}(\sigma)\boldsymbol{x} + c_{\text{out}}(\sigma)F_{\boldsymbol{\theta}}(c_{\text{in}}(\sigma)\boldsymbol{x}; \boldsymbol{c}, c_{\text{noise}}(\sigma)) \qquad (3)$$

Here, $F_{\boldsymbol{\theta}}(\cdot)$ represents the original output of the diffusion decoder. In the experiment, we used the magnitude and direction of vehicle speed as the target for generation.

**Experiment Setting.** We trained our diffusion model on the Waymo Open Motion Dataset (WOMD) (LLC, 2019). Each traffic scenario in the dataset has a duration of 9 seconds. We used the map information and the historical state of the previous 1 second as input to the model and generated future vehicle action sequences for the next 8 seconds. The training was conducted on a server with $4 \times$ Nvidia 4090 GPUs. We set the batch size for training to 16 and trained with the OneCycleLR learning rate schedule for 200 epochs. The entire training process lasted approximately 20 days. The detailed parameters of the model and the training process are shown in Table 5 and Table 6.

### A.4 GUIDE FUNCTIONS

Following (Zhong et al., 2023; Jiang et al., 2023), we calculate the cost function $\mathcal{L} : \mathbb{R}^{A \times T_f \times N_a} \mapsto \mathbb{R}$ based on the intermediate results of the generation process and propagate gradients backward to guide the final generation outcome. In our experiments, the control objectives include the vehicle's maximum acceleration, target velocity, time headway, and relative distance to the preceding car during car-following, and generating adversarial behavior by controlling nearby vehicles to approach the current vehicle. Denote vehicle $i$ at timestep $t$ has states $acc_{i,t}, v_{i,t}, x_{i,t}, y_{i,t}, heading_{i,t}$, and $dis_t(i,j)$ computes the relative distance between vehicle $i$ and vehicle $j$ at timestep $t$ when vehicle $i$ is followed by vehicle $j$ on the same lane. Table 7 shows the details of the cost functions.

Table 7: The cost functions used in the guided generation process.

| Guide Target | Cost Function |
|---|---|
| max acceleration | $\sum_{i=1}^{A} \sum_{t=1}^{T_f} \max(0, |acc_{i,t}| - acc_{max})$ |
| target velocity | $\sum_{i=1}^{A} \sum_{t=1}^{T_f} \| v_{i,t} - v_{target} \|_2^2$ |
| time headway | $\sum_{t=1}^{T_f} \sum_{i \neq j} \| \frac{dis_t(i,j)}{\|v_{j,t}\|_2^2} - thw_{target} \|$    where $i$ is followed by $j$ at $t$ |
| relative distance | $\sum_{t=1}^{T_f} \sum_{i \neq j} \| dis_t(i,j) - dis_{target} \|$    where $i$ is followed by $j$ at $t$ |
| goal point | $\sum_{i=1}^{A} \sum_{t=1}^{T_f} \| (x_{i,t}, y_{i,t}) - (x_{goal_{i,t}}, y_{goal_{i,t}}) \|_2^2$ |
| no collision | $\sum_{t=1}^{T_f} \sum_{i \neq j} \mathbb{I}[\| (x_{i,t}, y_{i,t}) - (x_{j,t}, y_{j,t}) \|_2^2 \leq \epsilon]$ |
| no off-road | $\sum_{i=1}^{A} \sum_{t=1}^{T_f} \mathbb{I}[\| (x_{i,t}, y_{i,t}) - (x_{\text{off-road}}, y_{\text{off-road}}) \|_2^2 \leq \epsilon]$ |

During the guided generation process, we use an Adam optimizer for the inner iterative gradient descent of Algorithm 1, we set learning rate $\alpha = 0.1$, clip parameter $\beta = 0.015$ and guide steps $K = 20$. For realistic guidance, the scale parameters for "no-collision" and "no off-road" are 12.0 and 2.5.

## B SIMULATION SYSTEM

### B.1 SCENARIO GENERATOR

We defined a unified traffic scenario format based on Protobuf[4]. Additionally, we have developed format conversion tools designed for the Waymo and Argoverse datasets, the conversion results can be seen in Figure 11. The detailed process of data construction can be found in our code base [5].

---

[4]https://anonymous.4open.science/r/LCSim-0C7A/lcsim/protos
[5]https://anonymous.4open.science/r/LCSim-0C7A/lcsim/scripts/scenario_converters

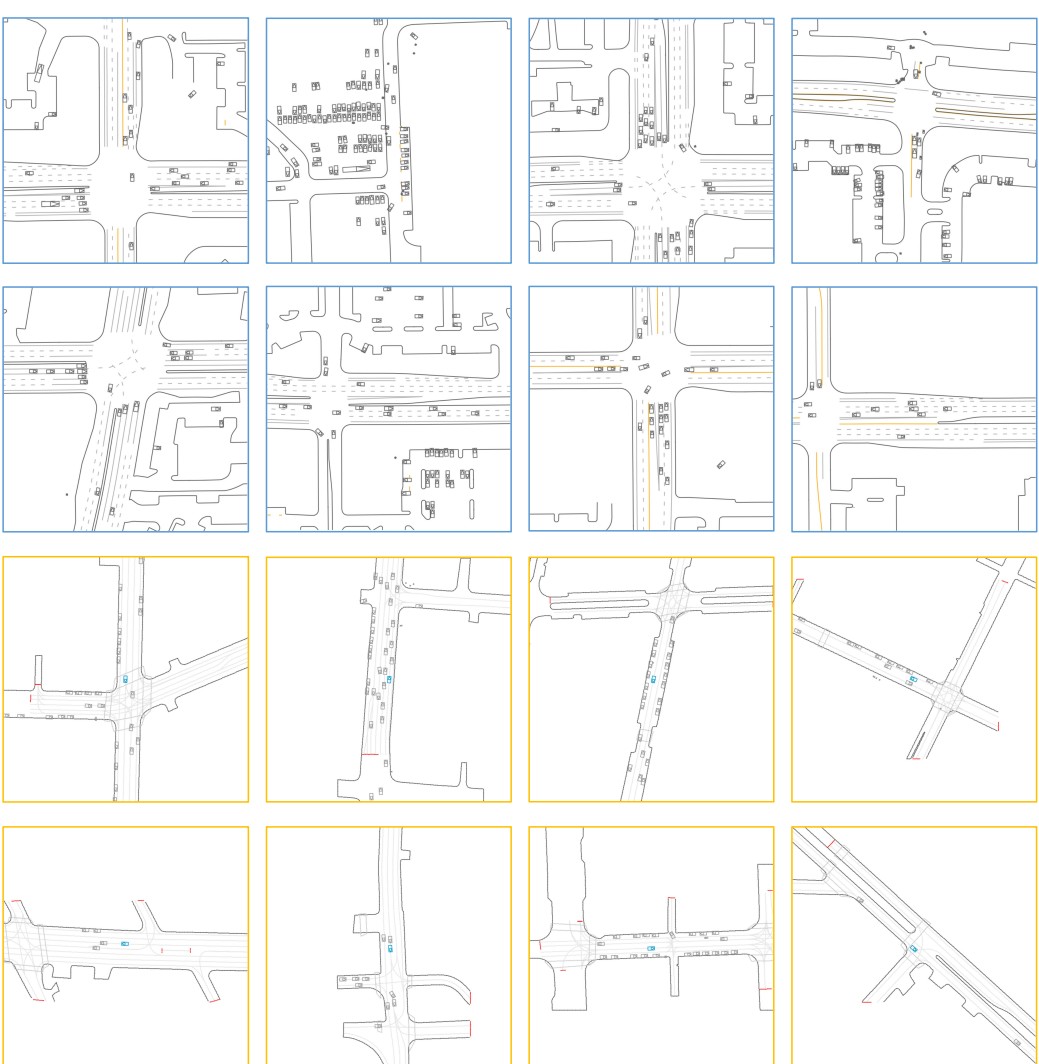

Figure 11: Traffic scenarios from WOMD (blue box) and Argoverse (yellow box).

## B.2 POLICY DETAILS

We implemented five different policies to support traffic simulation in various scenarios:

- *ExpertPolicy*: The vehicles strictly follow the given action sequences to proceed.

- *BicycleExpertPolicy*: Based on the expert policy, we impose kinematic constraints on the vehicle's behavior using a bicycle model to prevent excessive acceleration and steering. By default, we set max acceleration to $6.0\ m/s^2$ and max steering angle to $0.3\ rad$.

- *LaneIDMPolicy*: Under this policy, vehicles ignore the action sequences and proceed along the center line of their current lane. The vehicle's acceleration is calculated using the IDM model and lane-changing behavior is generated using the Mobil model.

- *TrajIDMPolicy*: Vehicles move along the trajectories computed based on the action sequence, but their acceleration is controlled by the IDM Mode to prevent collisions.

- *RL-based Policy*: A PPO (Schulman et al., 2017) agent trained based on our simulator, its observation space contains the scene embedding and the action sequence. The action space consists of acceleration and steering values. The training environment of this agent is the second one, enabling diffusive simulation with Waymo-style vehicle behavior.

For the IDM model in these policies, the default configuration is that $acc_{max} = 5m/s^2, thw = 2.0s, v_{target} = 20m/s$.

## C  PRIVATE DRIVING DATASET

### C.1  DATASET OVERVIEW

Our private driving dataset comprises about 426.26 hours of vehicle driving logs collected from autonomous vehicles in the Beijing Yizhuang area and the whole dataset is split into 765 scenarios. The data is presented in a format similar to vehicle trajectories in the Waymo dataset with a sampled rate of 10 Hz. However, the road networks of the scenarios are not provided in this dataset, so we can not train our model on it, but due to the sufficient duration of the data, we can analyze the behavioral characteristics of vehicles within the data collection area. This analysis provides a reference for constructing driving scenarios with different styles.

Understandably, due to confidentiality regulations, the complete dataset cannot be released. However, we will share the statistical distribution data of vehicle behaviors obtained from the dataset.

### C.2  VEHICLE BEHAVIOR ANALYSIS

We conducted statistical analysis on the dataset, focusing on metrics such as max acceleration, usual brake acceleration, velocity, relative distance, relative velocity, and time headway during the car following process, Figure 12 shows the results. This analysis allowed us to derive the driving behavior characteristics of vehicles in the Yizhuang area.

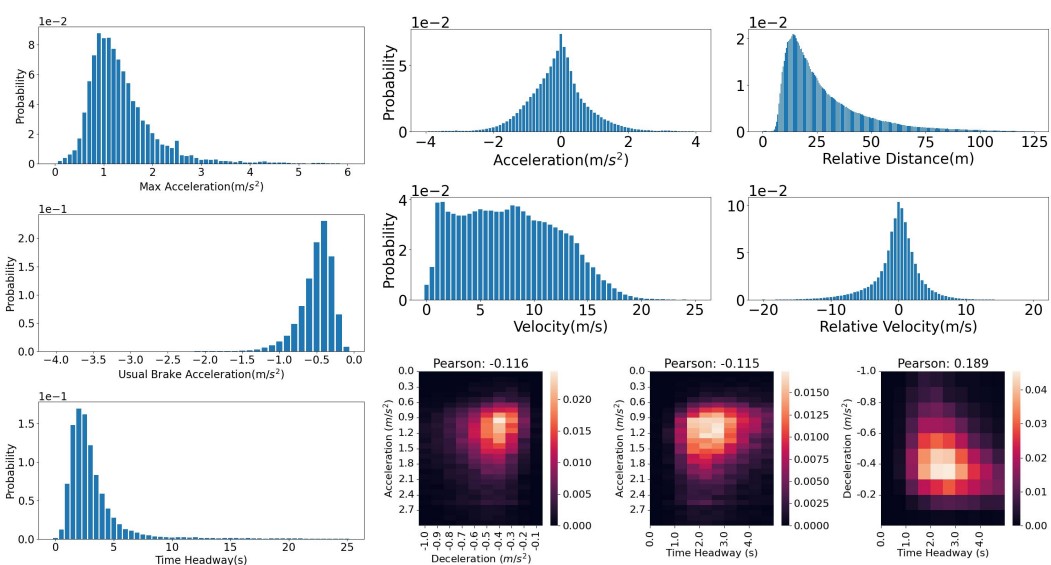

Figure 12: The analysis of our private driving dataset.

## D  MULTI-STYLE REINFORCEMENT LEARNING

We constructed single-agent reinforcement learning experiments based on the Waymo traffic scenarios with our guided diffusive simulation to see the influence of styles of scenarios on policy learning.

### D.1  REINFORCEMENT LEARNING SETUP

We constructed a reinforcement learning environment based on the validation set of the Waymo dataset. 4,400 scenarios are selected from the validation set and further divided into a training set

containing 4,000 scenarios and a test set containing 400 scenarios. We trained a PPO (Schulman et al., 2017) agent on the training set and evaluated its performance on the test set.

**Observation Spec.** Observation of the agent consists of two parts:

- Scene Embedding: Embedding computed by scene encoder of the diffusion model with size of $[N_h]$, by applying cross attention to map polygons and agent states, this feature contains information about surrounding vehicles, road elements, and the vehicle's own historical states. In this experiment, we use $N_h = 128$ following the setup of the diffusion model.

- Route: We sampled the vehicle's trajectory points within the next 1 second at a frequency of 10Hz and projected them into a relative coordinate system based on the vehicle's current position and orientation. The shape of the route data is $[10, 2]$, representing the reference path of the vehicle's forward movement. If the vehicle behavior in the driving environment is generated by a diffusion model, then this path will be accumulated from the behavior sequences generated by the model for the vehicle.

**Action Spec.** We let the agent directly control the throttle and steering angle of the vehicle. The agent's output is a two-dimensional vector with a range $[-1, 1]$. This vector is multiplied by the maximum range of acceleration and steering angle, resulting in the final vehicle action. In this experiment, the maximum acceleration and steering angle of the vehicle are set to 6.0 and 0.3, respectively.

**Rollout Setting.** To let the agent explore every scenario in the training set, we randomly divided the 4000 scenes in the training set into 20 parts, each containing 200 different scenarios. We used 20 parallel threads to rollout episodes, with each thread pre-loading and pre-calculating map embeddings for 200 different training scenarios. During the rollout process, after the current episode ends, the environment automatically switches to the next scenario, and this cycle continues iteratively.

**Reward Function.** Our goal is to make the vehicle progress along the given route while avoiding collisions and staying within the road. Therefore, we provide the following formula for the reward:

$$R = R_{forward} + P_{collision} + P_{road} + P_{smooth} + R_{destination}. \tag{4}$$

The meanings of elements in the formula are as follows:

- $R_{forward}$: A dense reward to encourage the vehicle to drive forward along the given route. We project the current position and last position of the vehicle onto the Frenet coordinate of the route and calculate $d_t, d_{t-1}, s_t, s_{t-1}$, the value of the reward would be $0.1 \times ((s_t - s_{t-1}) - (d_t - d_{t-1}))$.

- $P_{collision}$: Penalty for collision, When the vehicle collides, the value will be $-10$, and the current episode terminates; otherwise, the value is 0.

- $P_{road}$: Penalty for driving off the road, when this happens, the value will be $-5$, and the current episode terminates; otherwise, the value is 0.

- $P_{smooth}$: Following (Li et al., 2024), we implemented $P_{smooth} = min(0, 1/v_t - |a[0]|)$ to avoid a large steering value change between two timesteps.

- $R_{destination}$: When an episode ends, we check if the vehicle has reached the destination of the given route, which means the distance to the endpoint of the route is within 2.5 meters. If yes, the reward value is 10; otherwise, it's $-5$.

### D.2 MULTI-STYLE ENVIRONMENTS BUILDING

We build four kinds of environments with different driving styles using cost functions in Table 7:

- The original Waymo driving environment, in this environment, vehicles base their actions on real trajectories from the Waymo driving logs.

- The Waymo-style environment with diffusive simulation. This environment utilizes the diffusion model without guide functions, the vehicle behaviors are consistent with the Waymo dataset. With the diffusion model's nature, it generates diverse vehicle trajectories under the same initial conditions, exposing the agent to a broader range of traffic scenarios during training.

- The gentle-style environment with guided diffusive simulation. This environment follows the driving style observed in our private driving dataset, emphasizing a more "gentle" driving behavior compared to the Waymo-based environment. In this environment, we use cost functions on max acceleration with $acc_{max} = 3m/s^2$, and on time headway with $thw_{target} = 2.5s$.
- The adversarial environment. This environment is implemented by guiding nearby vehicles closer to the vehicle controlled by the RL agent. For vehicles in front of or alongside the main vehicle, we guide their action generation with $dis_{target} = 0$ to the main vehicle, thereby encouraging more sudden braking and cutting-in behaviors, increasing the aggressiveness of the environment.

## E DISCUSSION ON SIMULATION EFFICIENCY

As we mentioned before, efficiency is one of the main limitations of our simulator, here we compare LCSim and two baseline traffic simulators, MetaDrive and Waymax, we run scenarios in the validation set of WOMD with a length of 9s at 10Hz and compute the average simulation time per scenario. Table 8 shows the result. We use an Intel(R) Xeon(R) Platinum 8462Y CPU and an NVIDIA GeForce RTX 4090 GPU for the simulation. And LCSim can achieve comparable results with other cpu-based simulators implemented in Python. Our next plan is to develop a C++ version of our simulator.

Table 8: The attention mechanisms of scene encoder.

| Simulator | Metadrive | Waymax CPU | Waymax GPU | LCSim w/o Diff | LCSim w/ Diff |
|-----------|-----------|------------|------------|----------------|---------------|
| Time (s) | 1.923 | 0.554 | 0.083 | 0.239 | 1.043 |