# OpenReview forum: "LCSim: A Large-Scale Controllable Traffic Simulator"
_ICLR.cc/2025/Conference — ICLR 2025 Conference Withdrawn Submission_

### Official Review · Reviewer_172N · 2024-10-30

**Soundness:** 3
**Presentation:** 3
**Contribution:** 3
**Rating:** 8
**Confidence:** 5

**Summary:**

This paper proposes a new simulator in which diffusion policies control the background agents. One benefit is that we can adjust the background agents' behaviors while keeping their realism. The experiments are solid and demonstrate the effectiveness of this simulator in simulating traffic and training RL agents. As a system paper, the delivered software is of good quality with documentation.

**Strengths:**

**Impact**

Traffic simulation is important for the self-driving industry. Though real-world data helps improve the realism of background agents, none of the existing simulated agents can be controlled to show specific behaviors, driving styles, or even out-of-distribution ones that don't exist in the training data, i.e. collision. The use of diffusion in recent years shows the promise to solve this problem. Though using diffusion to generate traffic agents is not new, this work is the first one to try building a simulator with diffusion-based background agents and yields some new conclusions.

**Soundness**

The experiments are comprehensive and show some interesting conclusions:
1. The generated agents' trajectories match the distribution of the Waymo Dataset (test) very well
2. By adding guidance, the generated agents' collision/off-road rate can be further improved
3. DiffusionAgents trained with Waymo data can match the distribution of another dataset collected in Beijing by only adjusting the guidance function (without retraining), which is amazing.
4. Adjusting the guidance of background agents can produce different styles of traffic. Training single-RL agent in different styles of traffic produces different RL agent behavior. For example, RL agent will be conservative when the surrounding diffusion agents are with a guidance to drive close to them.
5. City-level traffic simulation shows the model can capture the traffic feature for different cities in terms of arrival time error, which further confirms its ability to model the data distribution.

Given the results, I believe the claim of a "controllable and realistic simulator" is well-supported.

**Quality**

As an open-sourced machine learning system work, the code/documentation quality of this work is good and easy to follow.

**Weaknesses:**

1. More qualitative results can be provided, especially the adversarial attack ones. It is tricky to generate reasonable adversarial behaviors rather than stupid rear collisions which is unavoidable for the ego car.
2. The conclusion of Table 3 seems to have Incorrect references, i.e. the third one is "with adv" not "with gentle".
3. What is the decision interval or simulation frequency? Would the closed-loop simulation performance change accordingly if the decision interval is changed? It seems to be an important hyperparameter for closed-loop simulation in the Waymo Sim Agent paper.
4. Is it possible to test open-sourced driving software in this simulator, like autoware? If it can be used to find misbehavior in open-sourced AD systems by adjusting background agents' behaviors, its impact can be enhanced.

**Questions:**

1. Is PCA used to do dimension reduction like MotionDiffuser?
2. Are all trajectories generated jointly instead of doing ego-centric motion prediction for all agents one by one?
3. The diffusion policy can be used to control not only the background agents but the ego car as well like what the RL agent did. Did the authors observe some interesting differences between the RL policy and diffusion policy when using it to control the ego car? Maybe the RL agents incur less infractions, since it is trained in closed-loop but the diffusion agent is basically doing behavior cloning?

---

### Official Review · Reviewer_D18U · 2024-11-01

**Soundness:** 3
**Presentation:** 4
**Contribution:** 2
**Rating:** 3
**Confidence:** 4

**Summary:**

The authors present a data-driven traffic simulator LCSim that has the characteristics of being large-scale, able to assimilate diverse data sources and perform controllable actions. LCSim takes as input a scenario and first inputs the map information and the historical information of all the agents into a scene encoder. The diffusion decoder then is responsible for providing future T_f steps of the agents in the scenario. The diffusion model uses a guided loss function with the following loss components: max acceleration, target velocity, time headway, relative distance, goal point, no collision, no off-road. The authors show comparisons between a private driving dataset and WOMD dataset and also the private dataset and the simulation dataset. The histograms show that the simulation can more closely follow the private dataset. The authors also trained a few RL agents on the simulation and provided metrics of the trained agents. The authors also provided city-scale scenario simulations.

**Strengths:**

- The diffusion-based motion planner for agents trained on WOMD provides a scalable and flexible way to model future actions of the agents
- The guided loss function provides a stable method for driving behavior of the agents
- The results provided are pretty comprehensive

**Weaknesses:**

- Three contributions are mentioned and I am not sure all of these are novel - First contribution is a unified data format for traffic scenarios. The authors do not mention OpenScenario which has been used as an unified data format. Also the third contribution is just mentioning the benchmarking the authors did for the paper, which does not count as a novel contribution.
- The purpose of creating an unified data format is vague and the different sources were not used in the later motion planning part.
- The diffusion-based motion planner is trained on WOMD - no reason is given why it is chosen and why it fares well when tested against a dataset collected in China.

**Questions:**

- The guide function penalizes and provides soft constraints but it cannot guarantee safety - are there any hard safety constraints that need to be provided?
- Line 183: "all map information is placed within a single junction" - What is the reason?
- Algo 1: t_i is increasing, but line 7 checks for t_{i+1} not equal to 0. Will never be 0.

---

### Official Review · Reviewer_5XAX · 2024-11-01

**Soundness:** 3
**Presentation:** 3
**Contribution:** 2
**Rating:** 5
**Confidence:** 4

**Summary:**

This work proposes LCSim, a large-scale controllable traffic simulator. The authors define a unified data format for traffic scenarios and tools to construct them from various data sources, e.g. WOMD and Argoverse. Furthermore, they design the simulator as well as a diffusion-based vehicle motion planner, which enables realistic and diverse traffic behaviors. The simulator provides an OpenAI Gym interface for RL and the authors use it to train RL agents with different driving styles and evaluate agents trained on WOMD and their own private dataset and compare them to each other.

**Strengths:**

* Simulator that supports several data formats: The authors developed a simulator based on a unified data format that supports several datasets that are otherwise not directly compatible to each other.
* Diffusion model beats baselines: The diffusion motion planner model beats TrafficSim and SimNet baselines.
* Dataset comparison: An interesting comparison between behaviors in WOMD and private datasets with differences in agent driving styles is provided.
* Well written: The paper is generally well written and easy to follow.
* Experiment details: The appendix contains relevant experiment details.

**Weaknesses:**

* Lack of significant novelty: Various simulators for traffic simulation exist (as shown in Table 1). The paper argues to develop a new simulator with a unified data format. However, instead of unifying all simulators it risks developing yet another simulator (https://xkcd.com/927/). The diffusion-based motion policy also lacks significant novelty and is similar to CTG.
* Single-threaded, slow simulator: Compared to Waymax (thank you for providing this comparison!), LCSim is slow and not accelerator-compatible.
* Simulator comparison: The simulator comparison in Table 1 does not seem totally fair. For example, Waymax does support data-driven and controllable agents, even though it does not provide one out of the box. Hence, the last sentence in Related Work that LCSim presents the first open-source traffic simulator with a controllable learning-based vehicle model is also too strong.
* Lack of related work: The unified data format that is provided is not compared to other formats, e.g. ASAM OpenDrive, that try to provide data format standards.
* Lack of strong baselines for diffusion planner: The baselines are both from 2021.

**Questions:**

* Open source code: Thank you for providing a link to the open sourced code + demos. I have tried to open the link but it would not load. Can you verify that everything works as intended?
* Baselines for diffusion planner: The baselines are both from 2021. Are there more recent methods this could be compared to? Can the motion planner also be used for sim agents and evaluated on the WOMD sim agent challenge?
* Difficult to read Figures 4, 5, 6: Orange + blue and orange alone look very similar and differences in the distributions are difficult to discern visually. Can you make these more readable?
* Lack of related work: How does your unified data format compare to ASAM OpenDRIVE?

---

### Official Review · Reviewer_A3Kp · 2024-11-04

**Soundness:** 2
**Presentation:** 2
**Contribution:** 1
**Rating:** 3
**Confidence:** 4

**Summary:**

This paper introduces LCSim.The authors claim it as a large-scale controllable traffic simulator designed to address the growing need for robust benchmarking of autonomous driving algorithms.

LCSim's key innovations lie in its unified data format for constructing traffic scenarios from multiple sources and its integration of a diffusion-based vehicle motion planner.

Authors claim the simulator supports large-scale traffic simulation by combining data from various datasets like Waymo, Argoverse, and OpenStreetMap, while offering controllable vehicle behavior modeling through guide functions. LCSim provides a Gym-like environment interface for reinforcement learning and can simulate different driving styles and behaviors.

**Strengths:**

- The integration of a diffusion-based motion planner allows for more realistic and diverse vehicle behaviors compared to traditional rule-based models.
- The unified data format for scenario construction from multiple sources (including Waymo, Argoverse, and OpenStreetMap) is a contribution, potentially enabling more comprehensive and varied simulation environments.
- The authors' implementation of guide functions for controlling vehicle behaviors adds a layer of flexibility that could be valuable for testing autonomous driving algorithms under different conditions.

**Weaknesses:**

My main concern lies in some overclaims and potential misrepresentations.

1. Firstly, the authors repeatedly refer to LCSim as a "simulation system" in both the title and contributions. However, a true traffic simulator should support "*closed-loop*" operations where vehicles run according to generated trajectories and continuously replan as the scenario changes. Despite the authors' claims, neither the manuscript nor the anonymous website demonstrates actual simulation effects, only showing one-time multi-vehicle joint  motion planning. If LCSim is indeed a simulator, the authors should particularly showcase experiments and results of continuous simulation over an extended period in a closed-loop setting.

2. Secondly, the authors emphasize that LCSim is "large-scale" and present experiments in Section 4.4. However, the Method section lacks explanation on how LCSim supports city-scale simulation, such as handling trajectory generation for hundreds or thousands of vehicles simultaneously, or supporting city-level road network operations and construction. The proposed diffusion-based motion planner seems unable to generate trajectories for so many vehicles in a city simultaneously, raising several unaddressed technical questions.
Section 4.4 only presents some city road network renderings and figures of road speed and arrival times, further adding to the confusion about how the authors implemented details such as "travel schedule of each agent in LCSim with a series of trips between intersections with corresponding departure time and arrival time" (line 466).

3. Lastly, even if considered as a multi-vehicle planner/predictor work, this paper *does not achieve state-of-the-art performance*. The two methods compared in Table 2 are from 2021 and outdated. When considering recent works from the Waymo SimAgent challenge (https://waymo.com/open/challenges/), such as BehaviorGPT (Collision 0.74, Off-Road 0.90) and MPS (Collision 0.96, Off-Road 0.93), we see that they significantly outperform LCSim (Collision 4.118, Off-Road 1.518).

**Questions:**

See weakness part, I hope the authors can calrify these issues.

---

### Note · Authors · 2024-12-14

I have read and agree with the venue's withdrawal policy on behalf of myself and my co-authors.